# Gene Expression of the D-Series Resolvin Pathway Predicts Activation of Anti-Tumor Immunity and Clinical Outcomes in Head and Neck Cancer

**DOI:** 10.3390/ijms23126473

**Published:** 2022-06-09

**Authors:** Domenico Mattoscio, Giulia Ferri, Claudia Miccolo, Susanna Chiocca, Mario Romano, Antonio Recchiuti

**Affiliations:** 1Department of Medical, Oral, and Biotechnology Science, University “G. d’Annunzio” Chieti-Pescara, 66100 Chieti, Italy; giulia.ferri004@studenti.unich.it (G.F.); mromano@unich.it (M.R.); 2Center for Advanced Studies and Technology (CAST), University “G. d’Annunzio” Chieti-Pescara, 66100 Chieti, Italy; 3Department of Experimental Oncology, IEO, European Institute of Oncology IRCCS, 20141 Milan, Italy; claudia.miccolo@ieo.it (C.M.); susanna.chiocca@ieo.it (S.C.)

**Keywords:** head and neck cancer, resolution of inflammation, resolvin D, human papilloma virus (HPV), pro-resolving lipid mediators, cancer immunotherapy, the cancer genome atlas database (TCGA)

## Abstract

In human medicine, the progression from early neoplasia development to either complete resolution of tumorigenesis and associated inflammation or chronicity and fatal outcomes remain difficult to predict. Resolution of inflammation is an active process that stimulates the termination of the inflammatory response and promotes return to homeostasis, while failure in resolution contributes to the development of a number of diseases. To understand how resolution pathways contribute to tumorigenesis, we defined and employed a cumulative score based on the expression level of genes involved in synthesis, signaling, and metabolism of the D-series resolvin (RvD). This score was used for comparative analyses of clinical, cellular, and molecular features of tumors, based on RNA-sequencing (RNA-seq) datasets collected within The Cancer Genome Atlas (TCGA) program. Our results indicate that higher RvD scores are associated with better clinical outcome of patients with head and neck squamous cell carcinoma (HNSC), and with molecular and cellular signatures indicative of enhanced anti-tumor immunity and better response to immune-checkpoint inhibitors (ICI), also in human papilloma virus (HPV) negative HNSC subtypes. Thus, higher activity of the RvD pathway identifies patients with improved resolution and a more efficient immune reaction against cancer.

## 1. Introduction

Acute inflammation is a beneficial immune response aimed to restore tissue integrity after pathological injuries. However, if not properly regulated, a chronic inflammatory state may ensue and drive a number of diseases including cancer [1]. Indeed, a large body of evidence has demonstrated that chronic inflammation provides growth, survival and pro-angiogenic factors, extracellular matrix-modifying enzymes, and mutagenic chemicals that fuel tumor development and progression [2]. Therefore, the identification of cancer patients with detrimental inflammation is of paramount importance.

Resolution represents the ideal outcome of acute inflammation and is an active process that encompasses the removal of the causative agent, a temporal shift in in the composition of the cellular infiltrate, i.e., from polymorphonuclear neutrophils (PMN) to monocytes-macrophages that, in turn, initiate the non-phlogistic phagocytosis of dangerous agents and apoptotic PMN [3]. These histological hallmarks of resolution, known by pathologists since the 19th century [1], are biochemically orchestrated by specific chemical lipid mediators called specialized proresolving lipid mediators (SPM) [4]. The SPM genus encompass four main superfamilies of bioactive compounds coined lipoxins (LX), resolvins (Rv), maresins (Mar) and protectins (PD), that differ in their biosynthetic process, chemical structure, and cognate receptors that convey biological actions [5]. The pioneering discovery of SPM represents a steppingstone for the comprehension of the pathophysiological mechanisms underlying many widely recurrent human diseases associated with unwanted chronic inflammation and for the generation of innovative concepts in pharmacology.

SPM are mainly produced by the actions of 15-LOX (*ALOX15* gene), 12-LOX (*ALOX12*), or 5-LOX (*ALOX5*) on arachidonic acid (LX) or docosahexaenoic acid (D-series Rv, protectin, maresin) (reviewed in [5]). The D-series Rv superfamily include six members of chemical mediators (RvD 1-6) which vary in the number, position, and chirality of their hydroxyl residues, as well as the position and cis-trans isomerism of their six double bonds. RvD are produced from docosahexaenoic acid (DHA) through 15-lipoxygenase (LOX)-mediated oxygenation at carbon-17, followed by a further oxygenation step catalyzed by 5-LOX. RvD modulate leukocytes infiltration and activities, and cytokines release to promote resolution of inflammation [6] via the activation of cell surface G-protein-coupled receptors (GPRCs). In humans, RvD1 and RvD3 activate ALX/FPR2 (*FPR2*) and DRV/GPR32 (*GPR32*). DRV/GPR32 was also reported as a receptor for RvD5, while RvD2 conveys pro-resolving functions through DRV2/GPR18 (*GPR18*) (reviewed in [7]). Finally, RvD endogenously produced during resolution of inflammation were further enzymatically converted to inactive lipids by 15-prostaglandin dehydrogenase (15-PGDH, *HPDG*) and the eicosanoid oxidoreductase Prostaglandin Reductase 1 (*PTGR1*) [8].

We and others recently reported pre-clinical evidence that the stimulation of resolution is beneficial to counteract primary and metastatic neoplasia [9,10,11,12,13]. However, if and how resolution pathways are defective in cancer and influence tumorigenesis is still poorly understood. To address this issue, we defined and employed a cumulative RvD score, based on the expression levels of genes involved in synthesis, signaling, and metabolism of RvD. This score was used as an estimator of pathway activity in comparative analyses, similarly to a previous report [14], to evaluate the clinical and molecular features of patients with head and neck squamous cell carcinoma (HNSC) within bulk RNA-sequencing (RNA-seq) datasets retrieved in The Cancer Genome Atlas (TCGA). We chose this model because chronic inflammation is a driving force in HNSC [15,16,17,18,19] and also because HNSC proved one of the most immunosuppressive microenvironments, despite a robust intratumoral immune infiltration, suggestive of failure in resolution [20,21].

Here, we show that higher RvD scores identify HNSC patients with improved clinical outcome, and with a molecular signature indicative of enhanced anti-tumor immunity and response to immune-checkpoint inhibitors (ICI). Thus, higher activity of the RvD pathway is associated with better immune reaction, improved resolution, and beneficial response to cancer.

## 2. Results

### 2.1. Definition of the RvD Score and Patients’ Stratification

To assess the relationship between the RvD score and clinical outcomes, RSEM-normalized gene expression values of TCGA HNSC bulk RNA-seq were mined from FireBrowse and used to calculate the RvD score as a surrogate biomarker of pathway activity, as in ref. [14], with minor modifications due to recent findings. This score accounts for the RvD production pathway, i.e., expression levels of *ALOX15* and *ALOX5* that work together and dependently (multiplication), with further dependence (multiplication) on receptors (*FPR2*, *GPR18*, or *GPR32* that work each other- addition), and inverse dependence (division) with inactivating enzymes (*HPGD* or *PTGR1* addition due to their independent actions) (Figure 1A). The final cumulative RvD score was thus calculated as shown in Figure 1B.

Tumor-specific HNSC RNA-seq datasets were then integrated with clinical features of matched patients, retrieved through cBioPortal [22] and stratified in high (the higher 100 tumor-patient pair) or low (the lower 100 tumor-patient pair) according to their cumulative RvD score. These subgroups were then used in comparative analyses to infer clinical outputs, differentially expressed genes (DEG) and biological functions in bulk RNA-seq, response to ICI, intratumoral immune cell infiltration levels, cell-type specific DEG and biological functions in major leukocyte subsets (Figure 1C). Furthermore, similar analyses were applied to human papilloma virus (HPV)-negative HNSC that generally has poorer prognosis than HPV positive HNSC [23].

### 2.2. High RvD Score Predicts Better Prognosis in HNSC

The expression levels of the distinct genes of the RvD pathway *ALOX15*, *ALOX15*, *FPR2*, *GPR18*, *GPR32*, *HPGD* and *PTGR1* were leveraged from 516 bulk tumor HNSC RNA-seq data, profiled by TCGA and plotted as a heatmap of individual tumor samples (Figure 2A). To evaluate the association between individual gene expressions of the RvD pathway and the survival time of patients, Cox proportional hazard models adjusted for age, gender and purity were used within the Gene outcome mode of TIMER 2.0 [24] (Figure 2B). As shown, the expression levels of the *GPR18* receptor gene is significantly associated with better prognosis in both HPV^+^ and HPV^−^ HNSC patients, thus suggesting the protective effect of signaling against cancer progression. Notably, *GPR18* expression is also associated with better prognosis in cervical cancer (CESC), another HPV^+^ tumor, as well as in different malignancies such as urothelial bladder carcinoma (BLCA), mesothelioma (MESO) and skin cutaneous melanoma (SKCM), further corroborating its tumor-protective role [25]. On the other side, *GPR32*, another RvD receptor, proved to be a worse prognosis index in other cancers such as kidney chromophobe (KICH), cervical kidney renal papillary cell carcinoma (KIRP) and in low grade glioma (TCGA-LGG). Similarly, other RvD receptors are associated with both protective and dangerous effects, depending on the cancer type. Conversely, the expression of the inactivating enzyme *PTGR1* is associated with an increased risk of death, in particular in HPV^−^ HNSC, while it is protective in kidney renal clear cell carcinoma (KIRC). Therefore, genes involved in the RvD pathway affect malignant progression in a cancer- dependent manner, thus underlying the importance of lipid mediators during tumorigenesis. 

To account for the complexity of RvD production and actions as a result of the concerted function of multiple enzymes and receptors (Figure 1A), HNSC patients were stratified according to the RvD score (Figure 1B) and analyzed for their clinical outcome. Patients with the higher RvD score showed significantly increased disease-free survival as compared to patients with the lower RvD score (Figure 2C), indicating that an elevated RvD score confers a clear survival benefit in HNSC (median months disease specific RvD high 210.97, RvD low 84.49). Consistent with this, patients alive or tumor-free dead are characterized by elevated RvD score in comparison to patients’ death with symptoms of cancer (Figure 2D), further corroborating that the RvD score correlates with HNSC clinical outcome. In contrast, the lipoxin score, calculated as (*ALOX5* + *ALOX12*) × (*ALOX15* + *ALOX5*) × (*FPR2* + *GPR32*)/(*HPGD* + *PTGR1*), applying the same principles of the RvD score, fails to predict HNSC survival (Logrank test *p*-value: 0.320) and clinical outcome (Figure 2E). Collectively, these results suggest that the RvD score is selectively associated with better outcomes in HNSC.

### 2.3. HNSC with High RvD Score Showed Transcriptional Fingerprints of Enhanced Anti-Tumor Immunity

To investigate the transcriptional landscape of tumors with the higher RvD score, bulk RNA-seq data of RvD high and RvD low tumors were compared at a single-gene level to identify DEG among groups. This analysis identified 254 down- and 2304 up-regulated transcripts (q-value < 0.05, fold change cut off ±2) in the RvD high group (Figure 3A,B). In addition, the Ingenuity Pathway Analysis (IPA) software revealed that genes modulated in the RvD high score group were associated with major biological themes related to anti-tumor immunity, such as Th1 pathway, and leukocyte activation and cytotoxicity, mainly driven by activation of interleukin-1β (*IL1B*), tumor necrosis factor α (*TNF*) and interferon γ (*INFG*) (Figure 3C). In accordance, gene ontology (GO) analysis with the Database for Annotation, Visualization and Integrated Discovery (DAVID) [26] confirmed that DEG in the RvD high group play a role in inflammation, immune response, cytokine and chemokine signaling, innate and adaptive immunity and T cell activation. Therefore, tumors with a higher RvD score display a transcriptional profile indicative of beneficial anti-tumor immunity.

To corroborate this finding, IPA was used to evaluate molecular, cellular and biological functions affected in tumors with a high RvD score (Figure 1C). At molecular and cellular levels, DEG in high RvD tumors are significantly associated with the regulation of cell-to-cell signaling and interaction, and cellular movement and signaling. At the functional level, DEG modulate the development and actions of blood cells, immune cell trafficking and immune response (Figure 4A). Among the most significantly affected functions, cell movement (Figure 4B) and activation (Figure 4C) of several leukocyte subsets are characterized by positive z-scores, indicative of increased activity. To infer infiltrated intratumoral leukocyte levels, bulk RNA-seq gene expression data of tumors with high vs. low RvD scores were analyzed by CIBERSORT (Figure 1C) to predict the abundance of 10 major leukocyte subsets (B cells, plasma cells, CD8^+^ and CD4^+^, T cells, NK cells, monocytes/macrophages, dendritic cells, mast cells, eosinophils, neutrophils). CIBERSORT effectively determines the overall immune content in RNA-seq datasets when compared against other experimental methods [27]. Tumors with a high RvD score displayed increased levels of anti-tumor effectors such as CD4^+^, CD8^+^ T lymphocytes (Figure 4D), and decreased infiltration of plasma and dendritic cells. In addition, IPA of DEG indicated that high RvD score tumors showed increased activation of both myeloid and lymphoid cells (Figure 4E), resulting in enhanced cell-mediated, cytotoxic, and humoral responses induced by the increased levels of immune-inflammatory mediators (Figure 4F). Collectively, DEG analysis of bulk RNA-seq data suggest that high RvD scores recognize HNSC tumors with altered immune effector dynamics and anti-tumoral activation states.

### 2.4. High RvD Score Identifies Potential Responders to ICI Therapies

ICI-targeting programmed cell death protein-1 (PD-1) and its ligands, programmed death ligand-1 (PD-L1)/2, rejuvenate antitumor immunity and produce clinical responses in subsets of HNSC, even if the majority of patients display primary resistance and do not benefit from these agents [28,29]. Appropriate patient selection is therefore crucial for the use of ICI. Increased levels of immune checkpoint targets, such as PD-L1 (*CD274*), Cytotoxic T-Lymphocyte Antigen 4 (*CTLA4*), Lymphocyte Activation Gene 3 (*LAG3*), Programmed Cell Death Protein 1 (*PDCD1*) may be used as biomarkers to identify putative ICI responsive patients [30,31,32]. Therefore, levels of *CD274*, *CTLA4*, *LAG3* and *PDCD1* bulk RNA-seq HNSC were retrieved and compared amongst RvD high and RvD low score patients (Figure 1C). This analysis revealed that tumors with a high RvD score exhibit significantly increased levels of all four ICI targets (Figure 5A,B), indicative of better responses to ICI therapies since pharmacological inhibition of these targets can re-enable effective immune attack. 

In addition, to evaluate if RvD scores are also correlated with immune activation, levels of end-products of anti-tumor immunity such as granzymes A and B (*GZMA* and *GZMB*, respectively), perforin (*PRF1*) and the cytolytic score [20,33] were evaluated in HNSC patients as surrogate markers of CD8^+^ T cell tumor cytotoxicity. RvD high tumors proved increased gene expression levels of anti-tumor effectors, including *GZMA*, *GZMB*, *PRF1*, and cytolytic score (Figure 5C–E), suggesting higher cytotoxic activity in the tumor microenvironment. Consistent with this, extended analysis of other targets for immunotherapy and markers of CD8^+^ T cell activation confirmed an up-regulation of most genes (e.g., *CD247*, *CD28*, *CD80*, *GSK3B*, *INFγ, LCK*, *IL12*, *PDCD1LG2*) involved in CD8^+^ T cell function in patients with high RvD score tumors (Figure 5F,G). Therefore, high RvD scores recognize HNSC tumors with elevated levels of immunotherapy targets and enhanced anti-tumor immunity, thus marking patients potentially responsive to ICI therapies. 

### 2.5. Immune Cell-Specific Gene Expression Profiles Revealed Increased Activation of Selected Leukocytes in Tumors with High RvD Score

To confirm that CD8^+^ T cells are hyper-reactive in RvD high-score tumors, the cell-specific transcriptomic profiles of selected leukocytes were inferred using CIBERSORT group mode [27]. To this end, bulk RNA-seq data from RvD high and RvD low score tumors were loaded on CIBERSORT and analyzed to determine the cell-specific gene expression of 10 leukocyte subsets (B cells, plasma cells, CD8^+^, and CD4^+^ T lymphocytes, NK cells, monocytes/macrophages, dendritic cells, mast cells, eosinophils, neutrophils) with LM10 as the signature matrix (Figure 1C). Then, the DEG for each leukocyte were calculated and plotted as a heatmap (Figure 6A). To characterize the immune landscape of single leukocyte populations, the DEG were analyzed on IPA. As expected, among the 10 immune-cell types analyzed, major differences in transcriptional profile were found on CD8^+^ T cells (Figure 6B) that showed activation of pathway (TCR receptor signaling and PD-1, PD-L1 cancer immunotherapy) and functions (immune response, expansion, function, development, quantity and cell proliferation) predictive of enhanced anti-tumor immunity in tumors with higher RvD scores. Along these lines, CD8^+^ T cells showed enhanced cytotoxicity in RvD-high tumors (Figure 6C), as also proved by the increased cytolytic score in cell-specific gene expression profiles (Figure 6D), similar to that observed in bulk RNA-seq (Figure 5E). Importantly, cell-specific expression analysis also confirmed that CD8^+^ T cells in high RvD score tumors express higher amount of genes for the inhibitory receptors CTLA4, LAG3 and PDCD1 (Figure 6E), indicating that these T lymphocytes could be efficiently unleashed by ICI, thus corroborating bulk RNA-seq results (Figure 5A,B).

In addition, PMN were also significantly affected in the HNSC RvD-high cohort, with increased activation of pathways related to differentiation, stimulation and immune response (Figure 6F). 

### 2.6. Higher RvD Scores Predicted Improved Anti-Tumor Immunity and Outcomes in HPV Negative HNSC

HPV negative HNSC show impaired prognosis and response to treatments compared to HPV positive HNSC [23]. Therefore, it was of paramount importance to determine if the RvD score identifies subsets of patients with superior anti-tumor immunity and increased survival in this cancer type. HNSC tumors were filtered according to their HPV status, as reported in cBioPortal, and compared for the RvD score. HPV negative HNSC exhibited lower RvD scores compared to HPV positive HNSC, thus suggesting that the poorer outcome of HPV negative patients could also, in part, be due to an impaired RvD pathway (Figure 7A). In addition, stratification of HPV negative HNSC patients according to their RvD score revealed that patients with the higher RvD scores have prolonged disease-free survival (210.97 vs. 84.49 months), also demonstrating the protective effects of the RvD pathway in this HNSC subset (Figure 7B). Importantly, gene expression analysis from bulk RNA-seq data evidenced increased levels of immunotherapy targets and effectors of CD8^+^ T cell antitumor immunity in HPV negative HNSC with a high RvD score (Figure 7C,D). Hence, high RvD scores are associated with improved outcomes and enhanced CD8^+^ T cell anti-tumor immunity, suggesting that stimulation of the RvD pathway may prime leukocyte-dependent inhibition of HNSC cancer progression.

To validate these findings, we determined whether RvD could prompt the anti-tumor activity of human leukocytes against HPV-negative cancer. To this end, blood borne human leukocytes were treated with vehicle or RvD1 (as a prototype of RvD with an increased score in the biobank atlas) and added on top of UM-SCC-23 cells. The growth curve of cancer cells was monitored with a real-time impedance assay in a small-scale in vitro system, as shown in [9]. As shown in Figure 7E, exposure to RvD1 potentiated the leukocyte-induced reduction in cell growth of UM-SCC-23 cells, thus indicating that stimulation of the RvD pathway (i.e., increased RvD production) is beneficial in an experimental in vitro model of HPV negative HNSC. These findings were in agreement with our recently reported results showing the activation of anti-tumor immunity prompted by RvD1 in models of HNSC HPV^+^ tumorigenesis (UM-SCC-104 cells) [9]. Of interest, there was no difference in the growth of UM-SCC-23 treated with RvD1 in the absence of leukocytes compared to vehicle-treated tumor cells used as a control, further indicating that RvD1 host directed anti-cancer actions intervene in the immune cell-tumor crosstalk, as unveiled in the genomic analysis.

## 3. Discussion

In this study we defined and exploited a RvD score to stratify HNSC patients accordingly to their resolution pathway activity, using a bioinformatic approach to evidence clinical, cellular, and molecular differences among patient cohorts (Figure 1). 

Here, we show that HNSC patients with an elevated RvD score showed extended survival, enhanced immune response, mainly driven by CD8^+^ T cells, and increased expression of immunotherapy targets, predicting a better response to ICI treatment (Figure 8). Therefore, an impairment in the RvD pathway (biosynthesis and signaling) could be detrimental in cancer, since it is linked to poorer prognosis and associated with a transcriptomic landscape predictive of weak anti-tumor immunity. These findings are in agreement with previous results, demonstrating that RvD supplementation reduces cancer growth in experimental models of tumorigenesis [9,10,11,12,13].

Importantly, we also evidenced similar outcomes for the HPV negative HNSC tumors (Figure 7), a HNSC subtype characterized by poorer prognosis, minor response to treatments, faint levels of tumor-infiltrating lymphocytes [23] and lower RvD scores (Figure 7A) than HNSC driven by high-risk HPV infection. Therefore, it is conceivable that stimulation of the RvD pathway could promote anti-tumor immunity against HPV-negative subtypes, similarly to HPV-positive tumors, as we previously reported [9]. Indeed, exposure of leukocytes to RvD1 reduced the cell growth of HPV-negative tumor cells (Figure 7E), thus validating the notion that prompting the RvD pathway in immune cells could be beneficial against cancers. This is consistent with our recent results showing that RvD1 activates anti-tumor immunity in vitro and in vivo disease models [9], further corroborating our present findings inferred from evidence collected in patients during the natural evolution of the disease. Importantly, these in vitro results demonstrate that the activation of pro-resolving pathways with RvD could be used as a point-of-care system for determining the patient’s ability to control the tumor.

The exploration of large transcriptomic datasets from tumor specimens has become an approach frequently used to define the immune landscape of cancer [9,20,33,34,35], as well as for biomarker discovery and validation [36,37,38]. Therefore, despite the fact that direct evidence of the correlation between the RvD score and the production of RvD isoforms still needs to be determined, our approach to comprehensively evaluate the RvD pathway holds several advantages, since it: (1) could be easily applied to freely accessible datasets; (2) allows us to determine the RvD score in a large number of tumors and patients, with strong, reproducible and consistent results; (3) permits the evaluation of the RvD pathway in tumors of different origin and anatomic location; and (4) gives an in silico estimate of RvD production and signaling without recurring to complex and expensive methodologies such as liquid chromatography/mass spectrometry. 

Using this strategy, we highlighted several important cellular and molecular features that characterize tumors with a high RvD score. By using a bioinformatic approach (CIBERSORT) that proved comparable to immunohistochemistry [27], we determined leukocyte infiltration levels. Tumors with a high RvD score evidenced increased CD4^+^ and CD8^+^ T cell infiltration, key for anti-tumor immunity in HNSC [39,40], but also decreased levels of anti-tumoral effectors such as plasma and dendritic cells, even if a specific subpopulation of these cells may also have a pro-tumorigenic role [41]. Therefore, increased activity of the RvD pathway significantly impacts on immune cell dynamics within the tumor microenvironment, likely due to alterations in cytokine/chemokine production (Figure 3D). Consistent with this, DEG analysis confirmed that genes regulated in high RvD score tumors are mainly involved in biological processes such as movement and activation of leukocytes (Figure 4B,C), thus confirming that enhanced RvD pathway activity is associated with the modulation of immunity in HNSC. These findings are in agreement with previous experimental observations obtained both in vitro and in vivo models of cancer and chronic inflammatory diseases, reporting alterations in cytokine/chemokine levels in inflammatory milieu coupled with the modification of immune cell infiltration and activity after stimulation of the RvD pathway [9,42,43,44,45].

Importantly, we identified CD8^+^ T cells as the main affected cellular effector in high RvD tumors. Indeed, both bulk DEG (Figure 3C and Figure 4B,C,E,F) and cell-specific gene expression profiles obtained from deconvolution of RNA-seq TCGA data (Figure 6B–D), pointed to an increased amount and activation of these key antitumoral cells. In addition, HNSC with higher RvD scores display elevated levels of cytotoxic molecules (Figure 5C–F, Figure 6C,D and Figure 7C,D), suggestive of potentiated killing ability against cancer cells. CD8^+^ T cells are the most powerful mediators of the anti-cancer immune response, and their intratumoral infiltration has been associated with better prognosis in several solid tumors, including HSNC [46,47]. Therefore, their increased levels and activity in RvD high score tumors could be the driving force underlying the better outcomes of HNSC patients with higher RvD pathway activity (Figure 2C,D). 

CD8^+^ T cells also play a crucial role in the response to immunotherapies [48]. ICI (e.g., pembrolizumab and nivolumab, monoclonal antibodies blocking the immune check point PD-1 on T lymphocytes) aim to block suppressive immune receptors to invigorate CD8^+^ T cell response against tumors [49]. Anti-PD-1 inhibitors have been recently approved for the treatment of recurrent or metastatic HNSC, in addition to conventional chemotherapy or as monotherapy for patients with HNSC expressing higher intratumor levels of PD-L1 (combined positive score ≥ 1) [50,51]. However, only a limited percentage of patients benefit from these treatments [28,29,30,31], despite a robust intratumoral immune infiltration suggestive of a strong immunosuppressive HNSC milieu [20,21]. Strikingly, we found that HNSC tumors with elevated RvD activity showed up-regulation not only of PD-L1 (*CD274*) but also of other genes encoding inhibitory receptors on lymphocytes (CTLA4, LAG3, PDCD1) (Figure 5A,B, Figure 6E and Figure 7C,D). Furthermore, RvD high tumors also express increased levels of other transcripts, such as *CD28* and *CD80*, which potentiate T cell receptor signaling, and intracellular molecules (*GSK3B*, *INFγ*, *LCK*, *IL12)* that stimulate immune responses. Unleashing the potential of CD8^+^ T cells in high RvD HNSC by ICI may therefore activate a potent cytotoxic reaction against cancer cells. Stratification accordingly to their RvD score may thus identify HNSC patients with a potential better response to ICI, thus allowing a specific tailoring of therapies only to responsive patients, with a considerable advantage in terms of the limitation of side effects and social costs. 

In addition to CD8^+^ lymphocytes, our results revealed a crucial impact of the RvD pathway activity in modulating PMN functions. Robust PMN infiltration in cancer tissues has been frequently associated with worse clinical outcome, although PMN can act as both pro-tumoral and anti-tumoral effectors depending on their phenotype [52]. PMN infiltrated in high RvD score HNSC showed markedly different transcriptional profiles than PMN in low RvD score tumors, mainly due to the activation of pathways involved in the modulation of the immune response (Figure 6F). In addition, to underline the key role of PMN in HNSC, this evidence confirms our recent experimental findings obtained in vitro and in vivo models of tumorigenesis showing the activation of anti-tumor immunity by RvD1-stimulated PMN [9]. 

Collectively, these results suggest that the stimulation of the RvD pathway (i.e., increased RvD production) could be beneficial to orchestrate the anti-tumoral activity of immune cells. Along these lines, our previous studies proved that RvD modulates the macrophage phenotype and controls the production of cytokines/chemokines/growth factors with pro-tumor properties [42,43,44]. Furthermore, other reports underline that RvD have potent actions on different immune cells including macrophages [10,11], NK [53] and T cells [10,54]. Hence, RvD polarize leukocytes towards an immune-active, pro-resolutive phenotype as part of their mechanism of action. Given these pleiotropic effects, RvD could represent ideal candidates to boost anti-tumor immunity that can be harnessed for therapeutic purposes.

## 4. Materials and Methods

### 4.1. Human Cancer Datasets and Clinical Analysis

The HNSC RNA-seq data (illuminahiseq_rnaseqv2 RSEM_genes_normalized) were downloaded from FireBrowse (firebrowse.org, accessed on 1 April 2022). Clinical data of Head and Neck Squamous Cell Carcinoma (TCGA, PanCancer Atlas) were retrieved from cBioPortal (cbioportal.org/datasets, accessed on 1 April 2022) [22]. Samples barcoded as normal or metastatic, and samples without information on clinical follow-up or survival were excluded. Of the 523 samples, 516 primary tumors including matched clinical and expression data were used for the analyses. Samples were ranked according to their RvD score, as described below, or filtered according to their survival status or HPV presence (415 HPV negative and 72 HPV positive).

Gene expression heatmaps from bulk RNA-seq were built through cBioPortal and reported as mRNA expression z-scores relative to diploid samples (RNA Seq V2 RSEM).

Cox proportional hazard model in the Gene_Surv module of TIMER2.0 (timer.cistrome.org, accessed on 1 April 2022) was used to evaluate the association between lipid gene expression and clinical outcome, adjusted for age, gender, stage and purity.

Survival curves were estimated by the Kaplan–Meier method using the survival module in cBioPortal after stratification of patients accordingly to their RvD score.

### 4.2. Definition of RvD Score

The HNSC RNA-seq data were used to filter out 11 genes currently identified as key in D-series resolvin pathway (*ALOX5, ALOX12, ALOX15, FPR2, GPR18, GPR32, CMKLR1, HPGD, PTGR1, PTGS1, PTGS2*). Similarly to [14], these genes were combined to calculate a RvD score that accounts for RvD production (*ALOX5* and *ALOX15*), RvD receptors (*FPR2, GPR18, GPR32*) and enzymes involved in RvD degradation (*HPDG* or *LTB4DH*). The final score was calculated as follows (genes involved in production x sum of gene receptors / sum of degradation genes):RvD score = *ALOX5* × *ALOX15* × (*FPR2* + *GPR18* + *GPR32* + *CMKLR1*)/(*HPGD* + *PTGR1*)
where multiplication is used for enzymes that work together, addition for independent or alternative enzymes, division for inverse dependence.

### 4.3. Identification of Differentially Expressed Genes and Pathways in Bulk RNA-seq HNSC Dataset

HNSC TCGA samples were ranked according to their RvD score and stratified in high RvD (the higher 100 samples) and low RvD (the lower 100 samples). Differential gene expression and statistical significance (q-value) among the two groups were calculated using the mRNA module of cBioPortal. mRNAs with a q-value < 0.05 and with a log2 fold-change (−log2 FC) ± 1 (corresponding to 2-fold change) were defined as differentially expressed. 

The functional association of genes and pathways selectively regulated in the high RvD group were identified by querying the list of differentially expressed genes in The Database for Annotation, Visualization and Integrated Discovery (DAVID) [26]. A false discovering rate (FDR) < 0.05 was considered significant. To infer the activation states (increased or decreased) of biological pathways modulated in the RvD high group, the activation z-scores and statistical significance (p value) of functions associated within the differentially expressed genes were calculated using the core analysis of Ingenuity Pathway Analysis software (IPA, Qiagen). Differentially expressed genes were defined as described above. Function and pathway analyses were retrieved by IPA filtering of experimentally observed relations in human tissues and primary cells using stringent filters. Pathways with a z-score > ± 2 and q-value < 0.05 were considered significant.

### 4.4. Profiling of Intratumoral Immune Cell Infiltration

To quantify intratumoral infiltration levels of immune cell fractions, bulk RNA-seq gene expression data from high and low RvD groups were analyzed with the impute cell fraction module of CIBERSORT [27], using LM10 as the gene signature matrix and disabled quantile normalization for RNA-seq data.

### 4.5. Leukocyte-Specific Gene Expression Profiles

To attribute cell-specific gene expression profiles, bulk RNA-seq gene expression data from high and low RvD groups were analyzed with the impute cell expression module, group mode of CIBERSORT [27]. For these analyses, the LM10 signature matrix was used to retrieve the gene expression profiles of 10 leukocyte subsets. Batch correction in B-mode and disabled quantile normalization were used to remove technical differences between the signature matrix and the input set of mixture samples. Datasets with a noise filter to eliminate unreliable estimated genes for each type were used to impute B cell, plasma cell, CD8^+^ T cell, CD4^+^ T cell, NK cell, monocyte, dendritic cell, mast cell, eosinophil and neutrophil gene expression profiles.

Differential gene expression between RvD high versus RvD low samples was calculated for each gene in paired leukocyte subsets and visualized as a heatmap using the BioVinci software. Genes with a −log2 FC > ± 2 in each population were considered differentially expressed and were analyzed with IPA to identify the pathways and biological functions significantly affected (z-score > ± 2 and q-value < 0.05). Function and pathway analyses were retrieved by IPA filtering experimentally observed relations in immune cells with stringent filters.

To impute CD8^+^ T cells-specific gene expression of *GZMA, PRF1, CTLA4, LAG3 and PDCD1* from bulk tissues at sample-level, a high-resolution module of CIBERSORT was used. Differential expression of selected genes in CD8^+^ T cells were then calculated with GraphPad Prism version 8.0.

### 4.6. Identification of Immunotherapy Responders

The gene expression levels of four key targets of immunotherapies [32] CD274 (Programmed death-ligand 1, PD-L1), cytotoxic T-lymphocyte-associated protein 4 (CTLA4), lymphocyte activation gene-3 (LAG3) and Programmed cell death protein 1 (PDCD1, PD-1) were retrieved through the mRNA module of cBioPortal in RvD high and RvD low tumor samples and visualized as a heatmap for their RNA-seq V2 RSEM expression values. The mean expression for each gene was calculated and plotted as a histogram using GraphPad Prism 8.0. Patients with higher immunotherapy targets were defined as immunotherapy responders [30]. Extended analysis of immunotherapy targets was carried out, retrieving the expression levels of genes involved in the PD-1, PD-L1 cancer immunotherapy pathway of IPA (*AKT1, AKT2, AKT3, B2M, BCL2L1, CD247, CD274, CD28, CD80, CDK2, CSK, CSNK2A1, CTLA4, FOXP3, GSK3B, HLA-DRB1, IFNG, IFNGR1, IFNGR2, IL12B, IL2RA, IL2RG, JAK1, JAK2, JAK3, LAG3, LCK, MR1, NGFR, PDCD1, PDCD1LG2, PIK3C3, PIK3CA, PIK3CB, PIK3CD, PIK3CG, PIK3R4, PRKCQ, PTPN11, RNF2, TGFB1, TGFB2, TGFB3, TNF, TNFRSF1A, TNFRSF1B and TYK2*) reported as mRNA expression z-scores relative to diploid samples.

Expression levels of genes encoding effector molecules released by cytotoxic CD8^+^ T lymphocytes (perforin- *PRF*, granzyme A- *GZMA*, granzyme B- *GZMB*) were used as markers of anti-tumor CD8^+^ T lymphocyte immunity as in [20]. RNA-seq V2 RSEM expression values retrieved through the mRNA module of cBioPortal in RvD high and RvD low tumor samples and visualized as a heatmap. The mean gene expression for each gene was calculated and plotted as a histogram using GraphPad Prism 8.0. As a further surrogate marker of cytotoxic activity, the cytolytic score was defined as the mean of *PRF1* and *GZMA* RSEM normalized values as in [33]. 

Tumor samples were also filtered according to their positivity to HPV mRNA and analyzed as described above to identify immunotherapy targets in HPV positive tumors.

### 4.7. In Vitro Cell Growth Analysis

For cell-growth studies, the University of Michigan Squamous Cell Carcinoma-23 (UM-SCC-23, target cells) [55] were maintained as we previously reported [56] and seeded at 25 × 10^4^ cell/well on iCelligence E-plate L8 (ACEA Biosciences) for 2 h before the addition of effectors. Buffy-coats from healthy donors were obtained from the ASL2 Lanciano-Vasto-Chieti (Abruzzo, Italy) transfusion center after informed consent. Human leukocytes (effector cells) were isolated after erythrocyte sedimentation in dextran (Merck KGaA) for 15 min at room temperature, washed, treated for 15 min at 37 °C with vehicle (0.05% Ethanol) or RvD1 (Cayman Chemicals) (10 nM) and added on top of UM-SCC-23 at an effector: target ratio of 10:1. Cell growth was monitored continuously up to 48 h with the real-time impedance assay (ACEA Bioscience) as in [9]. UM-SCC-23 treated with vehicle or RvD1 in the absence of leukocytes were used as the control.

### 4.8. Statistical Analysis

Statistical significance was evaluated with Graphpad Prism version 8.00. Multiple samples were compared with ANOVA. Paired or unpaired Student *t* tests were used to compare two samples normally distributed. Data were expressed as mean ± SEM of the number of biological replicates indicated in each figure legend. Values of *p* < 0.05 were considered statistically significant.

## 5. Conclusions

In summary, here we defined a RvD score to estimate D-series resolvin pathway activity and function in tumor specimen from the TCGA consortium. Our results demonstrate that HNSC patients, both positive and negative for HPV presence, with high a RvD score presented longer disease-free survival, enhanced anti-tumor immunity mainly driven by CD8^+^ T cells, and transcriptional profiles predictive of a better response to ICI treatment. 

## Figures and Tables

**Figure 1 ijms-23-06473-f001:**
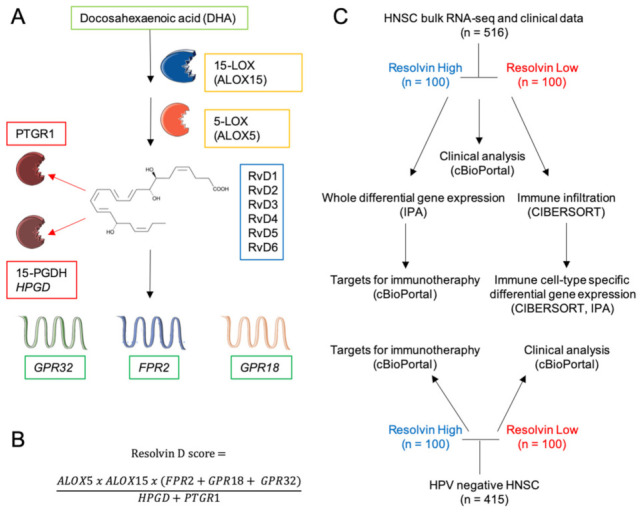
Schematics of Rv biosynthesis, score and experimental design. (**A**) Lipid precursor, biosynthetic and inactivating enzymes and receptors involved in D-series Rv pathway. The chemical structure refers to RvD1. The illustration was created with SERVIER Medical Art templates (smart.servier.com). (**B**) Equation for the calculation of the resolvin score. (**C**) Diagram of the experimental strategy and analyses.

**Figure 2 ijms-23-06473-f002:**
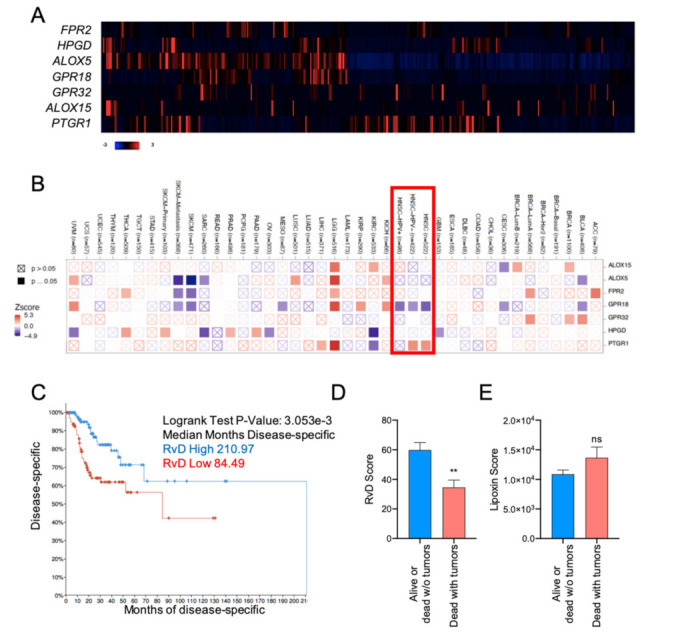
A higher RvD score is associated with more favorable outcomes in HNSC. (**A**) Heatmap of individual genes involved in SPM production in HNSC patients. mRNA expression z-scores relative to diploid samples (RNA Seq V2 RSEM) are shown. Patients are reported in columns, genes in rows. Blue: lower expression; red: higher expression. Color code is reported at the bottom of the panel. (**B**) Association between individual SPM genes and overall survival in multiple TCGA datasets. The heatmap (TIMER 2.0, Gene_Outcome mode) shows the normalized coefficient (z-score) of association between individual gene expression and overall survival (red: inverse association; blue: direct association) adjusted for age, gender, stage and purity. A red rectangle highlights HNSC cancers. Filled squares indicate significant (*p* < 0.05) associations predicted in multiple cancer types. TCGA study abbreviations were reported according to the NCI genomic data commons. (**C**) Kaplan–Meier curve survival analysis of HNSC patients stratified for high (n = 100) or low (n = 100) RvD score. Logrank test *p*-value and median months progression free are reported. (**D**) RvD score in HNSC from TCGA datasets in tumor-free alive or dead patients (n = 362) vs. patients’ death with symptoms of cancer (n = 128). Clinical data were extracted from the TCGA HNSC database using the cBioportal software. ** *p* < 0.01, Student *t* test. (**E**) Lipoxin score in HNSC from TCGA datasets in tumor-free alive or dead patients (n = 362) vs. patients dead with tumors (n = 128). Clinical data were extracted from the TCGA HNSC database using the cBioportal software. *p* = ns, Student *t* test.

**Figure 3 ijms-23-06473-f003:**
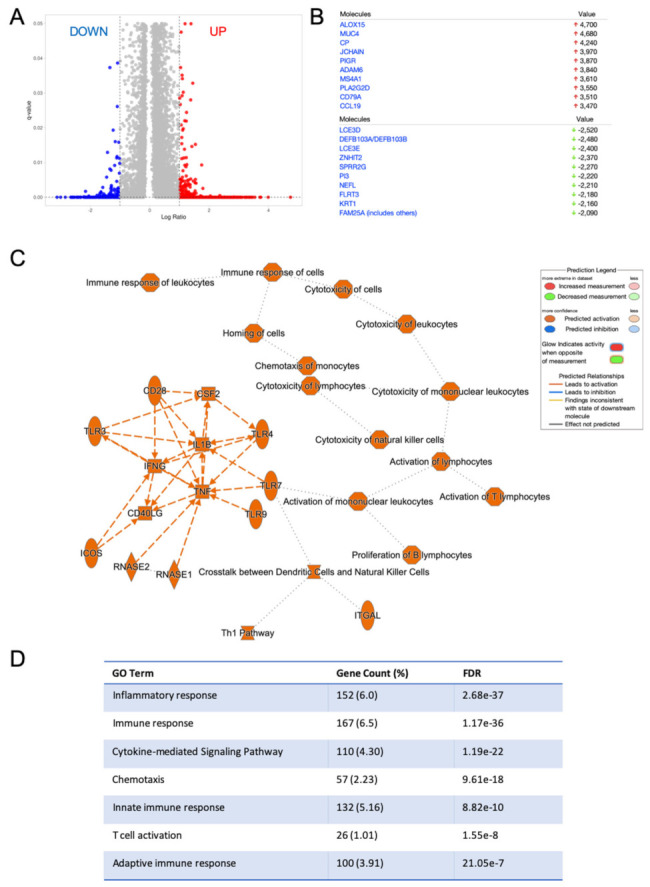
Differentially expressed genes in HNSC patients with a higher RvD score are associated with modulation of the immune-inflammatory reaction. (**A**) Volcano plot (prepared with VolcaNorseR) of gene expression patterns reporting up- down- (254 transcripts) or up- (2304 transcripts) regulated genes in patients with a high RvD score with a ±2 fold-change cut-off (corresponding to 1 in the log2 scale reported in the figure) and q-value < 0.05. (**B**) Top down (green arrows) and up-regulated genes in patients with a high RvD score. (**C**) Graphical summary of the major biological themes in high RvD tumors as determined by IPA core analysis of differentially expressed genes filtered as described above. The inset shows key legends. Solid lines represent direct interactions, while dashed lines represent indirect interactions. (**D**) Gene ontology (GO) of enriched biological themes in high vs. low RvD score patients identified by DAVID. Shown are GO Term, total and percentage of differentially expressed genes and False Discovery Rate (FDR).

**Figure 4 ijms-23-06473-f004:**
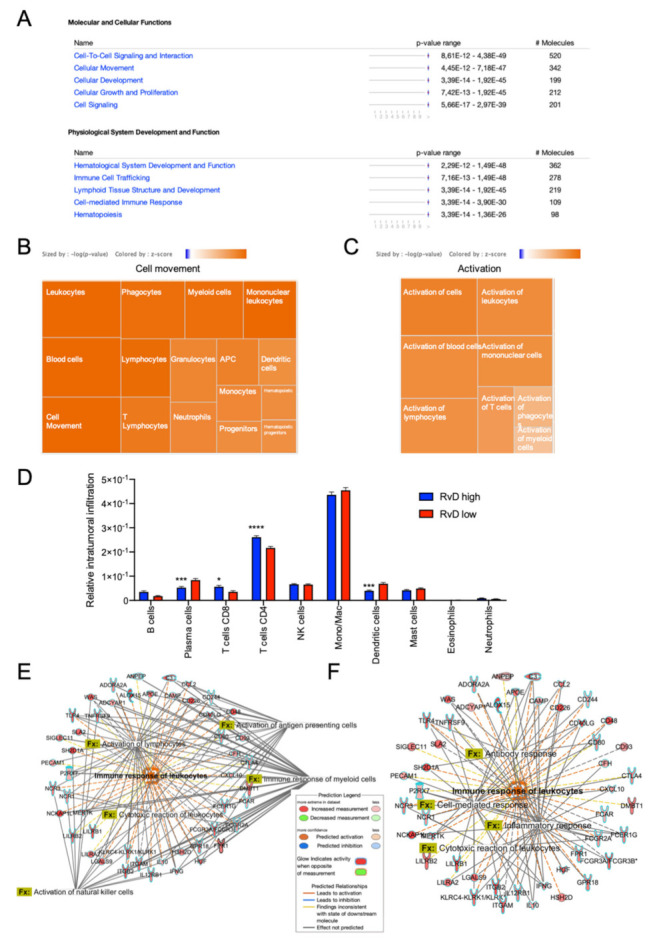
Anti-tumor immunity is boosted in patients with higher RvD scores. (**A**) IPA findings of biological functions associated with higher resolution scores. The most significantly affected molecular, cellular and biological functions and their relative *p*-value in patients with higher versus lower resolution scores are shown. (**B**) Activation z-score of biological functions related to cell movement. Each square reports the activation z-score (color legend is reported in the inset, sized by -log *p*-value) for the described biological functions. (**C**) Activation z-score of biological functions related to activation. Each square reports the activation z-score (color legend is reported in the inset, sized by -log *p* value) for the described biological functions. (**D**) Predicted tumor infiltration of 10 human immune subsets in HNSC samples with a high (n = 100) or low (n = 100) RvD score, as inferred by CIBERSORT. * *p* < 0.05; ***, *p* < 0.001; **** *p* < 0.0001, two-way ANOVA and Sidak’s multiple comparisons test. (**E**,**F**) Differentially expressed genes and significantly associated biological functions (Fx) related to activation (**E**) and response (**F**) of leukocytes.

**Figure 5 ijms-23-06473-f005:**
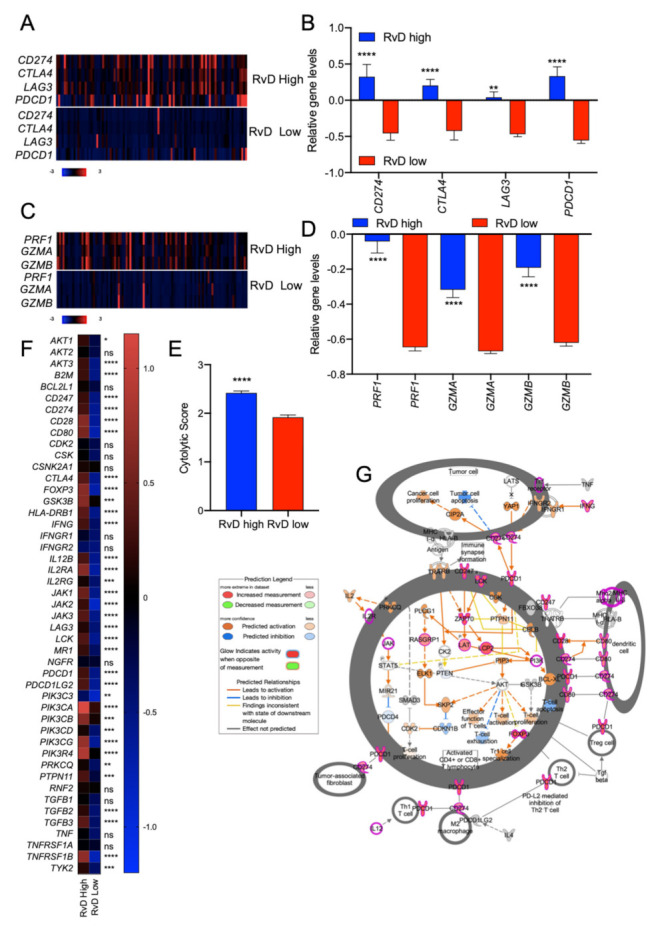
RvD score identifies putative immunotherapy-responder patients. (**A**) Heatmap of gene expression levels of four targets of immunotherapy (*CD274*, *CTLA4*, *LAG3*, *PDCD1*) in high (n = 100) vs. low (n = 100) RvD score HNSC tumors. mRNA expression z-scores relative to diploid samples (RNA Seq V2 RSEM) are shown. Patients are reported in columns, genes in row. Color code is shown at the bottom of the panel. Blue: lower expression; red: higher expression. (**B**) Mean gene expression levels (mRNA expression z-scores relative to diploid samples) of four immunotherapy targets in patients stratified as high (n = 100) and low (n = 100) accordingly to their RvD score. ** *p* < 0.01; ****, *p* < 0.0001, two-way ANOVA and Sidak’s multiple comparisons test. (**C**) Heatmap of gene expression levels of cytolytic compontents of effector CD8^+^ T cells in high (n = 100) vs. low (n = 100) RvD score HNSC tumors. mRNA expression z-scores relative to diploid samples (RNA Seq V2 RSEM) are shown. Patients are reported in columns, genes in row. Color code is reported at the bottom of the panel. (**D**) Mean relative gene expression (mRNA expression z-scores relative to diploid samples) of cytotoxic effectors in patients stratified as high (n = 100) and low (n = 100) according to their RvD score. ****, *p* < 0.0001, two-way ANOVA and Sidak’s multiple comparisons test. (**E**) Cumulative cytolytic score in high (n = 100) vs. low (n = 100) RvD score tumors. Data, shown as bars, are calculated on RSEM-normalized gene expression values. ****, *p* < 0.0001, Student *t* test. (**F**) Heatmap of expression levels of genes involved in the PD-1, PD-L1 cancer immunotherapy pathway (IPA). Mean expression levels of mRNA z-scores relative to diploid samples in tumors with RvD high (n = 100) and RvD low (n = 100) score are shown. * *p* < 0.05; ** < 0.01; *** *p* < 0.001; ****, *p* < 0.0001, two-way ANOVA and Sidak’s multiple comparisons test. (**G**) IPA representation of the PD-1, PD-L1 cancer immunotherapy pathway reporting up-regulation of immunotherapy targets and markers of anti-tumor immunity in T lymphocytes, fibroblasts, M2 macrophages and denditic cells.

**Figure 6 ijms-23-06473-f006:**
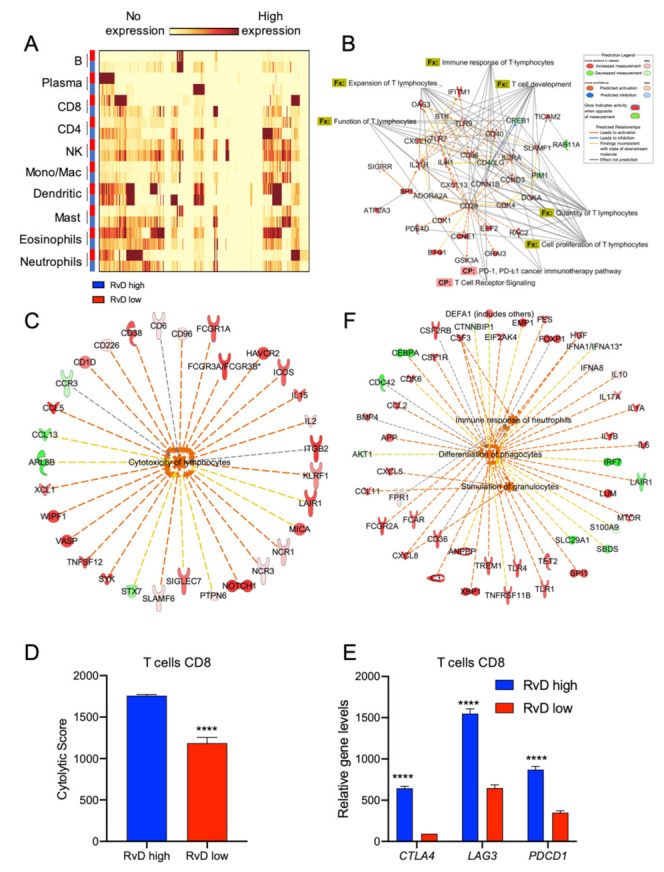
Higher RvD scores distinguish tumors with increased activation state in selected immune cells. (**A**) HNSC bulk RNA-seq were analyzed with the gene expression module of CIBERSORT to impute cell type specific expression profiles of 10 human immune subsets (LM10). The heatmap, generated with BioVinci, shows the gene expression profile of immune cells in tumors with high (blue rectangles) vs. low (red rectangles) RvD score. (**B**) IPA findings of biological functions (Fx) and canonical pathways (CP) associated with T lymphocytes activation and functions. (**C**) IPA findings of molecular network associated with increased lymphocyte cytotoxicity in RvD high score HNSC. (**D**) Cumulative CD8+ T cell cytolytic score in in high (n = 100) vs. low (n = 100) RvD score tumors. Data are shown as bars calculated on RSEM gene normalized values. ****, *p* < 0.0001, Student *t* test. (**E**) Mean gene expression levels (RSEM gene normalized values) of immunotherapy targets in CD8^+^ T cells stratified as high (n = 100) and low (n = 100) accordingly to their RvD score. ****, *p* < 0.0001, two-way ANOVA and Sidak’s multiple comparisons test. (**F**) IPA findings of biological functions associated with PMN activities.

**Figure 7 ijms-23-06473-f007:**
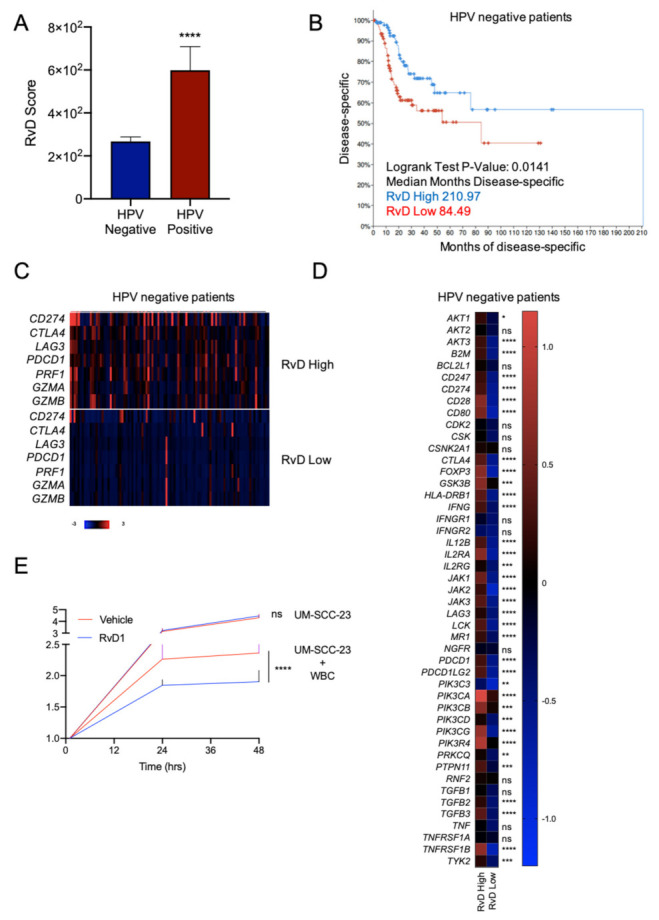
Higher RvD scores are associated with improved outcomes in HPV negative HNSC. (**A**) RvD score in HPV negative (n = 415) vs. HPV positive (n = 72) HNSC. **** *p* < 0.0001, Student *t* test. (**B**) Kaplan–Meier curve survival analysis of HPV negative HNSC patients stratified as high (n = 100) or low (n = 100) RvD score. Logrank test *p*-value and median months progression free are reported. (**C**) Heatmap of gene expression levels of immunotherapy targets and cytotoxic effectors in high (n = 100) vs. low (n = 100) RvD score in HPV negative HNSC patients. mRNA expression z-scores relative to diploid samples (RNA Seq V2 RSEM) are shown. Patients are reported in columns, genes in row. Color code is reported at the bottom of the panel. Blue: lower expression; red: higher expression. (**D**) Heatmap of gene expression levels of genes involved in the PD-1, PD-L1 cancer immunotherapy pathway (IPA). Shown are the mean expression levels of mRNA z-scores relative to diploid samples in HPV negative tumors with RvD high (n = 100) and RvD low (n = 100) score. * *p* < 0.05; ** < 0.01; *** *p* < 0.001; ****, *p* < 0.0001, two-way ANOVA and Sidak’s multiple comparisons test. (**E**) Growth of the HPV negative HNSC cell line UM-SCC-23 during co-incubations with purified blood-derived human leukocytes from n = 5 healthy donors exposed to RvD1 (10 nM) or vehicle control. Cell growth was monitored up to 48 h with an impedance-based real time cell analysis (ACEA). Target UM-SCC-23 cells treated with vehicle or RvD1 in the absence of leukocytes were used as control. Data are expressed as relative cell growth normalized at the start of treatments. ****, *p* < 0.0001, one-way ANOVA.

**Figure 8 ijms-23-06473-f008:**
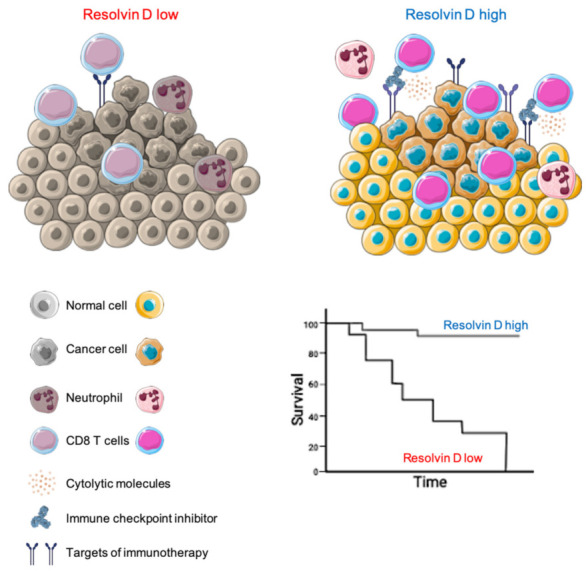
Proposed model of how the RvD pathway affects the tumor microenvironment to improve HNSC outcomes. The illustration was created using SERVIER Medical Art templates (smart.servier.com).

## Data Availability

Publicly available datasets were analyzed in this study. This data can be found here: cbioportal.org; firebrowse.org; timer.cistrome.org.

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
