# Peer review of "Gene Expression of the D-Series Resolvin Pathway Predicts Activation of Anti-Tumor Immunity and Clinical Outcomes in Head and Neck Cancer"

_ijms, 2022, doi:10.3390/ijms23126473_

Round 1

Reviewer 1 Report

This outstanding paper by Mattoscio et al describes an association between a higher RvD score and better clinical outcomes for patients with HNSC. There is a particularly strong prediction of improved outcomes for HPV-negative HNSC, as well as an improved response to ICI therapies. The results are convincing given their usage of large transcriptomic datasets from primary human tumors. Overall, they demonstrate the implications of higher RvD activity with potentially improved prognosis in HNSC.

Suggestions-

The authors state that that the poorer outcome of HPV negative patients [compared to HPV positive patients] could be also in part due to impaired RvD pathway – can you validate this with in vitro assays? Also, the in vitro finding only looked at HPV negative HNSC – it would be good to see it in HPV-positive HNSC as well.

They briefly touch upon how GPR32 is a worse prognosis index in certain cancers – expand upon why this may be?

They suggest in the discussion that the RvD pathway is affecting immune cell dynamics likely due to alterations of cytokine/chemokine production – can potentially do further experiments to demonstrate altered cytokine levels in vitro.

Minor

  • Double check that grammar and spelling is correct
  • Could the authors use different phrasing to refer to patients? “dead with tumor” seems too casual here.
  • Figure 2C axis legends are confusing – this is a survival curve, but what does “disease-specific” mean? Should it be progression-free survival?

Reviewer 2 Report

 Mattoscio et al. defined and employed a cumulative score based on the expression level of genes involved in synthesis, signaling, and metabolism of the D-series resolvin (RvD). This score was used for comparative analyses of clinical, cellular, and molecular features 22 of tumors, based on RNA-sequencing (RNA-seq) datasets collected within The Cancer Genome 23 Atlas (TCGA) program. This is an interesting study; I recommend a major revision before being considered for publication.

Here are my major concerns:

  • The authors claimed that “RvD score as surrogate biomarker of pathway activity as in ref. [14], with minor modifications due to 86 recent findings.” However, it is unclear what are these minor modifications? How they are doing better compared to the referenced one?
  • What is the reason to pick 100 samples each way in HNSC patients? Why are not the patients divided into two groups based on the median or mean of the score?
  • Can authors show the correlations between infiltrating immune cell population and the score using scatterplots colored by patient purity? Figure 4D can be biased by sample purity easily.
  • Can authors apply this score to blood samples from HNSC patients?

Round 2

Reviewer 2 Report

Authors addressed my concerns.